# Pancreatic Cancer Resistance to Treatment: The Role of Microbiota

**DOI:** 10.3390/biomedicines11010157

**Published:** 2023-01-07

**Authors:** Enrico Celestino Nista, Angelo Del Gaudio, Livio Enrico Del Vecchio, Teresa Mezza, Giulia Pignataro, Andrea Piccioni, Antonio Gasbarrini, Francesco Franceschi, Marcello Candelli

**Affiliations:** 1Medical and Surgical Science Department, Fondazione Policlinico Universitario Agostino Gemelli—IRCCS, Università Cattolica del Sacro Cuore, 00168 Rome, Italy; 2Emergency Medicine Department, Fondazione Policlinico Universitario Agostino Gemelli—IRCCS, Università Cattolica del Sacro Cuore, 00168 Rome, Italy

**Keywords:** pancreatic cancer, microbiota, therapy resistance, immunotherapy

## Abstract

Pancreatic cancer (PC) is an aggressive malignancy and the fourth leading cause of cancer death in the United States and Europe. It is estimated that PC will be the second leading cause of cancer death by 2030. In addition to late diagnosis, treatment resistance is a major cause of shortened survival in pancreatic cancer. In this context, there is growing evidence that microbes play a regulatory role, particularly in therapy resistance and in creating a microenvironment in the tumor, that favors cancer progression. The presence of certain bacteria belonging to the gamma-proteobacteria or mycoplasmas appears to be associated with both pharmacokinetic and pharmacodynamic changes. Recent evidence suggests that the microbiota may also play a role in resistance mechanisms to immunotherapy and radiotherapy. However, the interactions between microbiota and therapy are bilateral and modulate therapy tolerance. Future perspectives are increasingly focused on elucidating the role of the microbiota in tumorigenesis and processes of therapy resistance, and a better understanding of these mechanisms may provide important opportunities to improve survival in these patients.

## 1. Pancreatic Cancer

Pancreatic cancer (PC) is an aggressive malignant tumor and the fourth leading cause of cancer death in the U.S. and Europe. PC kills nearly 50,000 people each year in North America alone. In fact, PC has the lowest survival rates for all stages combined (11%) among malignancies, and although the incidence is relatively stable, it is estimated to be the second leading cause of death by 2030 [1]. Because the symptoms of PC are usually nonspecific, the disease is usually not diagnosed until the late stages, when the patient complains of abdominal pain, unexplained weight loss, fatigue, nausea, or jaundice. Among exocrine tumors of the pancreas, which account for 95% of PC, pancreatic ductal adenocarcinoma (PDAC) is the most common and accounts for almost all pancreatic malignancies [2]. PDAC usually arises from the neoplastic transformation of pancreatic intraepithelial neoplasms (PanINs) and intraductal papillary mucinous neoplasms (IPMNs), typically found in 2% of the population, and is more common in people older than 70 years [2]. Several risk factors are significantly associated with the development of PC. For example, advanced age (71 years), male gender, heavy alcohol consumption, obesity, low vegetable diet, chronic pancreatitis (CP), diabetes, smoking, and family history have been shown to increase the risk [3]. In particular, cigarette smoking appears to be the cause of nearly 20% of pancreatic tumors [1], and current and former smokers have a 1.56- and 1.15-fold increased risk, respectively, according to a recent meta-analysis by Ben QW [4]. Having two or more first-degree relatives with PC is also a relevant risk factor, and it is estimated that approximately 8% of cases of PC are diagnosed in patients with a positive family history [5,6]. According to various meta-analyses, the prevalence of diabetes has increased threefold in PC patients. Moreover, this association is even higher in patients recently diagnosed with diabetes, which could be an early symptom of cancer [7].

Another risk factor is chronic pancreatitis, and analysis by Gandhi et al. estimated the standard incidence ratio (SIR) to be 22.61 in patients with chronic pancreatitis and as high as 63 in hereditary types [8]. By contrast, numerous other factors, such as coffee consumption [9], oral contraceptive use [10], and proton pump inhibitor use (PPI) [11], which were considered relevant additional risks in the past, showed no statistical association in further analyses. Tumorigenesis results directly and indirectly from alterations in various molecular pathways and is correlated with different genetic alterations, such as mutations in TP53, KRAS, CDKN2A, and SMAD4 [6]. However, genetic testing is currently indicated only for certain populations. BRCA2 genetic testing is recommended in patients with a family or personal history of breast cancer, with a family history of PC, and in individuals of Jewish descent. The presence of the CDKN2A mutation should be tested in patients with a family history of hereditary melanoma [12]. In addition to clinical examinations, the diagnostic process provides a complete blood analysis to monitor the indices of cholestasis and the presence of acute pancreatitis. The carbohydrate antigen 19-9 (CA 19-9) remains the most used serum tumor marker for PDAC, with a sensitivity of 80% among symptomatic patients. CT scans of the abdomen have a crucial role in diagnosing and staging the tumoral tissue. In addition to the CT technique, MRI or PET could also be included during the staging process, mostly in case of indeterminate and occult lesions [3]. Eventually, because of the high sensitivity and minimally invasive technique, endoscopic ultrasound (EUS) represents the first-choice modality for obtaining biopsies and definitive diagnosis [3].

### 1.1. Therapies for Pancreatic Cancer

In the PDAC treatment spectrum analysis, surgery is considered the only curative treatment for this neoplasm, but only a small percentage (10–20%) of patients can benefit from this treatment, as more than half of patients have the metastatic disease [1]. Adequate staging is required to differentiate patients with potentially resectable, borderline resectable, locally advanced, and metastatic disease [13]. In the early stages of cancer, the initiation of a neoadjuvant therapy protocol before surgery is still controversial [14]. The American Society of Clinical Oncology (ASCO) guidelines recommend that neoadjuvant therapy be reserved for patients with suspected extra pancreatic disease or radiographic evidence of mesenteric vessel infiltration who are at high surgical risk and have high serum levels of CA 19-9 without cholestasis [15]. The most commonly used strategy for resectable disease is surgical resection followed by adjuvant therapy. In this context, recent National Comprehensive Cancer Network (NCCN) guidelines recommend gemcitabine alone, gemcitabine in combination with capecitabine, continuous fluorouracil infusion (5 FU), or 5 FU/leucovorin therapy (NCCN) [16]. In borderline resectable and locally advanced diseases, no further surgery is recommended by the NCCN. At this stage, the choice is between chemotherapy or chemoradiotherapy as neoadjuvant therapy. For this purpose, FOLFIRINOX (a combination of folinic acid, fluorouracil, irinotecan, and oxaliplatinum), gemcitabine-nab-paclitaxel, and gemcitabine-cisplatin are currently approved as chemotherapy [15,16]. As previously reported, most patients are diagnosed with advanced-stage metastatic disease, with a median one-year survival rate of 7% [17]. Therapeutic strategies for metastatic disease include Nab-paclitaxel plus gemcitabine, FOLFIRINOX, gemcitabine and capecitabine, gemcitabine and erlotinib, gemcitabine, and cisplatin, or gemcitabine alone [14]. In addition to these therapies, the current interest in elucidating the molecular mechanisms and biology of cancer has led to the development of new targets and new modalities for PDAC treatment [14], which are summarized in Table 1. A panoramic of the novel treatments in development for PC is presented.

### 1.2. Microbiota and Pancreatic Cancer

There is growing evidence on how the gut microbiome, the genome of the entire community of microorganisms living in the gastrointestinal tract, can influence human health and disease [6]. New detection methods such as whole genome sequencing have facilitated the description and detection of an increasing number of bacteria, fungi, and viruses that can influence host inflammatory status, intestinal permeability, and carcinogenesis [18,19]. Gastrointestinal tumors, such as pancreatic ductal adenocarcinoma (PDAC), have a higher number of detections for type-specific microbiome fingerprints, mainly due to their spatial proximity [6,20]. The reciprocal balance between the pancreatic and gastrointestinal microbiota can lead to various pancreatic pathologies [21]. While various metabolites of the microbiota, such as short-chain fatty acids (SCFA), protect against tissue inflammation, and normal pancreatic β-cells secrete cathelicidin-related antimicrobial peptides (CRAMP) control pancreatic bacterial overgrowth [22]. In recent decades, several studies have highlighted the association between intestinal and oral dysbiosis and the presence of PC [23,24,25,26,27,28,29,30,31] (Table 2).

Dental disease and associated changes in the oral microbiome increase the risk for PDAC, as evidenced by decreased oral concentrations of *Neisseria elongata* and *Streptococcus mitis* and the increased abundance of *Leptotrichia* and *Porphyromonas gingivalis* in patients with PC [24,25]. In recent decades, the presence of certain species, such as *P. gingivalis* or *Helicobacter pylori,* has been associated with higher rates of PC. Several pathological pathways have been proposed to explain this correlation. In particular, *P. gingivalis* may affect miRNA expression, immunological response, and arginine catabolism [26,27]. Furthermore, studies in animal models and in vitro have shown that *P. gingivalis*, *Treponema denticola*, and *Tannerella forsythia* can induce KRAS and p53 mutations and increase the risk of PC [28]. In addition to the luminal microbiota, other studies have demonstrated the physiological presence of a microbiome associated with pancreatic tissue. Specifically, bacterial cytidine deaminase (CDA) appears to be expressed four times more in PCs than in healthy controls [29], with a higher average abundance of the microbiome [18]. In general, the bacterial communities in the pancreas had a similar composition in terms of taxa as the oral microbiome. In addition, several studies have identified host variations in the genes involved in the immune response associated with different β-diversity [30]. Compared to healthy controls, PC patients showed an increased relative abundance of *Porphyromonas*, *Prevotella*, *Bifidobacterium*, and *Synergistetes*, as well as the phylum *Euryarchaeota* [18,31]. In another study, PDAC with KRAS mutations was associated with the enrichment of the genera *Acinetobacter, Sphingopyxis*, and *Pseudomonas*. PDAC tissues and fluid from IPMNs were enriched in *Fusobacterium nucleatum*, an oral pathogen [31], and depleted in *Lactobacillus* and phyla of *Firmicutes* and *Proteobacteria* [32], suggesting a role for Fusobacterium in the development of pancreatic neoplasia [33]. Bacterial communities may influence tumor progression through various metabolites such as SCFAs, gallic acid, lipopolysaccharide (LPS), and polyamines [34]. LPS is a common component of Gram-negative bacteria and could provide another pro-oncogenic stimulus by promoting local inflammation via the Toll-like receptor (TLR) [34,35] and subsequently mutating and activating the KRAS gene [34]. In addition, P53 mutations appear to be associated with a specific bacterial metabolite. In particular, gallic acid, produced by *Bacillus subtilis* and *Lactobacillus Plantarum*, was recently linked to pro-oncogenic Tpr53 mutations in a mouse model [36]. Numerous microorganisms can synthesize polyamines that are directly involved in cell growth [37]. These metabolites were significantly elevated in the mouse models of PC, and their reduction lead to a lower rate of cell anabolism [6,37]. In addition to bacterial communities, fungi have also been shown to be involved in the pathogenesis of PC [38]. Strikingly, the mycobiome of PDAC tissues is usually more pronounced, especially in the genera *Malassezia* and *Alternaria*, thus differing from the normal intestinal mycobiome. The pathogenic role of fungal components was mediated by the stimulation and secretion of IL-33 and activation of TH2 cells and ILC2. In mouse models, IL -33 was able to induce a chromatin switch that interacted with a genetic mutation such as KRAS [38]. In addition, glycans of fungal walls can also pursue oncogenic processes themselves by binding mannose-binding lectin (MBL) and recruiting the complement system [39]. Although antifungal treatments have appeared to halt carcinogenesis in previous in vitro experiments [40], they have not shown efficacy in animal models, likely due to the low concentrations of live fungi in tissues [41]. Several studies have found specific microbiome differences between patients with long-term survival (LTS) and short-term survival. In a study by Riquelme E et al., it was demonstrated that patients with a better prognosis had increased relative abundance of *Alkalihalobacillus clausii* (the new name of *Bacillus clausii*), *Pseudoxanthomonas*, *Saccharopolyspora*, and *Streptomyces*, with higher alpha diversity in their tissue-associated microbiome. Researchers transferring fecal microbiota transplantation (FMT) from LTS to mice also demonstrated the causal effect of these species in modulating tumor growth [40]. Moreover, the poorer prognosis was associated with the detection of intra-tumoral fusobacteria, although no specific differences in genetic mutations were found [23]. A subsequent analysis based on 187 PDAC patients also showed that certain strains were more prevalent in patients with metastases. Specifically, they found a higher abundance of *Beutenbergia cavernae DSM 12333*, *Mycoplasma hypopneumoniae*, *Tolypothrix sp. PCC 7601*, *Acidovorax ebreuus TPSY*, *Agrobacterium radiobacter K84*, *Aggregatibacter aphorophilus NJ8700*, *Shigella sonnei Ss046*, *Salmonella enterica*, *Citrobacter freundii* and the lower presence of *Saccharomonospora Viridis DSM 43017* [42]. Although further studies are needed to clarify the clinical role of gut microbiota in pancreatic cancer, a recent promising study has demonstrated the importance of fecal microbiota analysis in diagnostic evaluations [43]. The authors considered 27 different species, such as *Fusobacterium nucleatum/hwasookii*, *Alloscardovia omnicolens*, *Veillonella atypica*, *Faecalibacterium prausnitzii*, *Bifidobacterium bifidum Romboutsia timonensis*, *Bacteroides coprocola*, *Veillonella atypica*, *Methanobrevibacter smithii*, *Bacteroides finegoldii,* and *Alloscardovia omnicolens*, noted an ability to discriminate between the disease and health of 84%, and of 94% when combined with serum CA 19-9 [43].

### 1.3. Microbiota and Chemotherapy

Therapy resistance is one of the main causes of shortened survival in pancreatic cancer [41]. Among the various factors involved in resistance, the gut microbiota seems to play an important role, mainly affecting drug metabolism and absorption [44,45] (Figure 1).

Interestingly, some bacteria belonging to the gamma-proteobacteria class, such as *Escherichia coli* [46], can interfere with gemcitabine metabolism by converting this drug to its inactive form (2*′*,2*′*-difluorodeoxycytidine to 2*′*,2*′*-difluorodeoxyuridine) via their long isoform of the bacterial enzyme CDA [29].

In addition, gamma-proteobacteria were detected at elevated levels in human PDAC. (29) Furthermore, other genera, such as Mycoplasma, can produce cytidine deaminase, and the use of tetrahydrouridine (THU), an inhibitor of CDA, can restore the normal anticancer effects of gemcitabine [47]. These effects have been extensively studied in colorectal cancer, and there are emerging data on PC. For example, colon cancer cells were found to develop chemotherapy resistance when cultured with bacteria derived from human pancreatic cancer, which are normally enriched in proteobacteria [48]. Moreover, in a recent study, the presence of K. pneumonia in the bile ducts was associated with a worse prognosis in the population treated with adjuvant therapy and gemcitabine, and a higher survival rate was observed after antibiotic therapy with quinolones [49]. On the other hand, the increased microbial production of vitamin K2 (menaquinone), usually detected in diabetes models, has also been associated with resistance to gemcitabine/paclitaxel [50]. As a result, the hypothesis of anti-oxidative cell protection against the accumulation of reactive oxygen species induced by chemotherapy has been put forward [50]. Another drug class of interest for microbiota-mediated chemoresistance is fluoropyrimidines, which are included in the FOLFIRINOX protocol [51]. As mentioned above, some studies have shown an increased inactivation rate of 5 FU sustained by *Mycoplasma hyorhinis* [52] and by *Fusobacterium nucleatum (F. nucleatum)* through the alteration of TLR4/MYD88-dependent autophagic metabolism, and the inhibition of 5-FU-induced cell apoptosis. Furthermore, the abundance of *F. nucleatum* has been linked to poor prognosis in pancreatic cancer [40,53,54]. Although no specific enzyme or mechanism has yet been identified [29], *F. nucleatum* has also been associated with resistance to oxaliplatin [54]. By contrast, some species, such as *Bacteroides ovatus* and *Bacteroides xylanisolven*, have been associated with enhanced recruitment of T cells and, consequently, with the enhancement of the effect of erlotinib: an EGFR tyrosine kinase inhibitor [55].

### 1.4. Microbiota and Immunotherapy

Immunotherapy is considered an effective therapeutic strategy for many neoplasms, such as melanoma and non-small cell lung carcinoma [56,57]. Immunotherapies mainly use monoclonal antibodies to stimulate an anti-tumor immune response directed against programmed cell death protein 1 (PD-1)/programmed death ligand 1 (PD-L1) or cytotoxic T lymphocyte antigen 4 (CTLA-4). Unfortunately, these strategies have not yet proven effective in PDAC [58], likely due to the specific characteristics of the tumor microenvironment (TME). The complex organization of the extracellular matrix, fibroblasts, and immunosuppressive leukocytes creates a difficult-to-target environment that promotes tumor growth [59]. An important role in this complex mechanism of resistance is also attributed to the microbiota. Recent evidence has shown that microbes influence the response to immunotherapy, regulate immune checkpoints, and promote cancer cell escape from the immune system [60,61,62]. The use of antibiotics in PDAC mouse models confirmed these findings and led to profound changes in the immune system, such as an increase in effector T cells and a change in tumor-associated macrophages (TAM) [18,62]. Other relevant changes included a higher proportion of antitumor cell types, such as Th1 CD8, and the release of gamma interferon and TNF alfa, as well as a decreased distribution of Treg cells and the production of IL-17 [63]. Notably, bacterial suppression has been associated with the increased expression of PD-1 and increased efficacy of checkpoint-targeted immunotherapy, with a synergistic effect on tumor size and an increase in T-cell activity [18]. Countervailing evidence is that the effect of antibiotics on reducing tumor growth did not occur in Rag1 knockout mice lacking mature T (and B) lymphocytes. This suggests that the immune system and microbiota must be involved in this process and that it is not a direct cytotoxic effect of the drug [61]. Moreover, the microbiota can contrast the effect of antibiotics by blocking IL 17, which, together with CD4 and th17 cells, is an important pro-carcinogenic factor in TME. [61]. A relatively new approach to modulating the gut microbiota is FMT, and a better prognosis and response to immunotherapy was observed in PC animal models that received stools from healthy donors or long-term survivors (LTS). Conversely, the use of antibiotics after FMT from healthy controls did not affect survival [39]. Other elements that appear to play a role in the immune evasion of PDAC are microRNAs, which have also been associated with increased tumor growth and resistance to therapy. MicroRNAs have been shown to alter pancreatic cell gene expression and influence immune responses, and recent studies have linked some bacteria to the expression of such miRNAs, suggesting additional pathways of immune regulation [64]. In a recent study, it was confirmed that the presence of megasphaera, capable of producing SCFA in tissues, showed a better response to anti-PD-1 therapies [65]. At present, promising preclinical model results have been achieved from the combination of anti-PD1 and antibiotics. On the contrary, clinical trials to evaluate the combination of pembrolizumab (NCT03637803) with lyophilized bacteria, or the combination of probiotics, vancomycin, and nivolumab (NCT03785210), are still in progress [18].

### 1.5. Microbiota and Radiotherapy

Radiotherapy (RT) is a therapeutic strategy based on ionizing radiation that results in direct damage to DNA, the formation of reactive oxygen species (ROS), and reactive nitrogen species (RNS) [66]. Irradiation can induce an intense immune response leading to tumor cell death and the release of antigens capable of stimulating cells of the immune system, such as CD8 T lymphocytes, resulting in a significant antitumor effect capable of killing tumor cells in distant, non-irradiated parts of the organism. This effect is known as the abscopal effect [60]. RT has been shown to be useful in resected PDAC patients to improve local control as an adjuvant therapy due to its cytostatic activity [51]. However, although the techniques are based on irradiating the tumor areas as selectively as possible, there are also effects on the surrounding healthy tissue, especially considering the anatomical location of the pancreas. In this context, a complex relationship between the microbiota and the effect or tolerance of radiotherapy has been recently identified [67]. In some studies with germ-free mice, increased resistance to the radiation effect has been described, both in terms of the therapeutic effect and toxicity [68]. Conversely, an enhanced effect of RT was observed after the administration of vancomycin in cancer models. This synergistic result was attributed to the elimination of Gram-positive bacteria, which can interfere with the presentation of tumor antigens to cytolytic CD8+ T cells [69]. Further evidence for the role of the microbiota as a predictor of the response to RT is provided by hepatocellular carcinoma (HCC). A recent study found that c-di-adenosine monophosphate (c-di-AMP), a bacterial metabolite, enhances the RT-related immune response [70]. Although these studies were not performed in pancreatic cancer, they justify the hypothesis of the influence of the microbiota on the immune system during RT treatments and thus provide the basis for future correlation and causality studies in this neoplasm.

### 1.6. Bidirectional Relationship between Therapy and Microbiota

In recent years, a bidirectional relationship between microbiota and therapeutic strategies in PDAC has been established. Both chemotherapy and radiotherapy can lead to alterations in the gut microbiota, which may limit its tolerability and influence its toxicity [63]. As mentioned earlier, radiation can cause tissue damage leading to enteritis [71], and the gut microbiota appears to play a central role in these processes. The most commonly observed changes after irradiation are the decrease in *Lactobacillus* spp. and *Bifidobacterium* spp. and the increase in *Escherichia coli* and *Staphylococcus* spp. [72]. These changes lead to the impairment of intestinal barrier integrity and the production of proinflammatory cytokines. In this regard, pretreatment with FMT or an antibiotic cocktail to restore the intestinal microenvironment after irradiation provides interesting results [73]. As for traditional chemotherapy, several studies are based on gemcitabine therapy. During treatment, an increased abundance of proinflammatory bacteria, such as Proteobacteria, and a decrease in butyrate-producing bacteria, such as Lachnospiraceae and Ruminococcaceae, have been described. These changes are associated with an increase in intestinal permeability, inflammation, and a decrease in the anticancer activity of butyrate, which promotes tumor progression [74,75,76]. In addition, a decrease in SCAF production has been noted during paclitaxel therapy, which has been associated with a higher rate of *Clostridioides difficile* infection [77] and an increase in Mucispirillium in the colon, which may play a role in neuroinflammation and cause peripheral neuropathy [78]. With respect to the drugs included in the FOLFIRINOX protocol, toxicities related to changes in the microbiota have been described for 5-FU as well as for oxaliplatin and irinotecan. Therapy with 5-FU is frequently compromised by the occurrence of mucositis, likely due to the depletion of mucin-producing bacteria such as Lactobacillus and Streptococcus [79]. Therefore, it has been reported that the administration of Lactobacillus and Bifidobacterium could increase the production of anti-inflammatory cytokines and thus reduce the intensity of the damage [80]. The efficacy of oxaliplatin is often limited by gastrointestinal toxicity and pain; interestingly, these effects are reduced in germ-free mice after antibiotic administration or after FMT [81]. The mechanisms involved are drug-induced increased levels of bacterial products, such as LPS, in the spinal ganglion, leading to inflammation [71]. In addition, the role of the microbiota in irinotecan-mediated diarrhea has recently been investigated [45]. In particular, this therapy seems to increase the abundance of Enterobacteriaceae, leading to an increase in bacterial β-glucuronidase (BGUS) and LPS, both of which are responsible for inflammatory responses and intestinal damage. Accordingly, BGUS activity has recently been proposed as a predictive marker for irinotecan-induced diarrhea [82], and it has been reported that the co-administration of irinotecan with a selective inhibitor of bacterial β-glucuronidases can prevent irinotecan toxicity in mice [83].

## 2. Conclusions and Future Perspectives

As mentioned above, the microbiota-based approach is an important resource in the therapy of PDAC, as it influences the efficacy and tolerability of chemotherapy, immunotherapy, and radiotherapy. Accordingly, the modulation of the microbiota is one of the most studied topics at present. These strategies include the use of FMT, probiotics, prebiotics, and antibiotics. Riquelme et al. studied the role of FMT in the mouse models of pancreatic cancer. They showed a better clinical response at five weeks in mice that received stools from patients who had the disease more than five years after resection compared to mice that had advanced PDAC patients as donors. In addition, their study showed a higher infiltration of CD8+ T cells into tumor tissue in mice that received FMT from patients who had the disease more than five years after resection, demonstrating the role of their microbiome in modulating immune response and survival [40]. FMT may also have an important effect in reducing the toxicity of chemotherapy, as demonstrated in animal models, suggesting a role in improving treatment adherence and compliance [81]. Interestingly, this effect is also observed with radiotherapy. In mice, the recovery rate after radiation treatment was shorter in animals receiving FMT, likely due to the differential expression of long noncoding RNA by microorganisms [84].

Probiotics and molecules derived from them may also be effective against pancreatic cancer and significantly reduce infectious complications after pancreatoduodenectomy, according to a recent study [85]. Ferrichrome, a substance produced by *Lactobacillus casei* ATCC334, can inhibit pancreatic, gastric, and colon cancer cells [52,53]. Ferrichrome could induce the M1 phenotype of tumor-associated macrophages (TAMs) via a TLR4-dependent pathway, thereby increasing immune system activities and the efficacy of immune checkpoint blocker therapies [86,87,88]. The administration of next-generation probiotics, such as *Akkermansia muciniphila*, *Prevotella copri*, *Parabacteroides goldsteinii*, and *Faecalibacterium prausnitzii*, represents a novel strategy [89]. In particular, *A. muciniphila*, a Gram-negative bacterium that can support gut immunity and cytokine release, significantly inhibits the proliferation of rat pancreatic islet tumor cells [90]. Interestingly, in certain neoplasms, such as renal or pulmonary carcinoma, the presence of resident microbiota has proven necessary for the effectiveness of immunotherapy. In a different way, in pancreatic carcinoma, the presence of certain bacteria appears to be a factor against therapeutic effectiveness. Indeed, the study of the mechanisms of chemical resistance induced by bacteria led to the recognition of specific targets to antagonize this effect. This includes, for example, CDA inhibitors and inhibitors of the bacterial enzyme β-glucuronidase (GUS), the clinical value of which has to be determined in the future.

The use of antibiotics in PDAC, either in combination with chemotherapeutics or immunotherapeutics, is currently being tested or appears to shorten overall survival [91]. Another goal of antibiotic administration and microbiota modulation is to improve the tolerability of therapy. An example is the improvement of irinotecan-induced diarrhea and the demonstration of reduced intestinal toxicity in germ-free mice or mice treated with broad-spectrum antibiotics. 

On the other hand, it should be kept in mind that prolonged antibiotic therapies may lead to side effects, alter the balance between the host and microbiota in all body regions, and result in the selection of antibiotic-resistant bacterial species, so further studies are essential to evaluate the safety of such strategies. One area of great interest is the genetic manipulation of microbes with the aim of producing vectors that are capable of enhancing immune responses, spreading toxins, transferring DNA or RNA to specific targets, and altering the expression of oncogenes in cancer cells [92].

The main studies concern *Salmonella typhimurium* and *Listeria monocytogenes*, but human trials of this approach have yet to demonstrate superiority over conventional treatment. This requires more trials to better assess the clinical implication of this mechanism.

Finally, lyophilized probiotics and vancomycin may increase the efficacy of pembrolizumab [93] and nivolumab [94], although clinical trials have not yet been completed [95]. In summary, future perspectives are increasingly focused on clarifying the role of the microbiota in tumorigenesis and therapeutic resistance processes [96,97,98]. Combining microbiota modulation with conventional treatments such as chemotherapy, radiation, or immunotherapy seems to improve efficacy and tolerability and represents an important way forward. However, further studies specifically tailored to pancreatic cancer and performed in human models are needed for proper understanding and future clinical applications of this knowledge.

## 3. Materials and Methods

A review of the literature was performed to focus on the role of gut and tissue microbiota in influencing the effectiveness and metabolism of medical treatment for pancreatic cancer. The electronic databases that were searched included PubMed, Medline and Google Scholar, Scopus, and Embase. We focused on the following keywords and terms: “Pancreatic Cancer”, “Resistance”, Treatment”, “Therapy” AND “Microbiota”. We sought these terms within titles, abstracts, and keywords.

The studies included in this review were carefully reviewed by 2 authors. We included all papers with the full text available, original works, and metanalysis.

Exclusion criteria were language other than English and availability only of the abstracts.

## Figures and Tables

**Figure 1 biomedicines-11-00157-f001:**
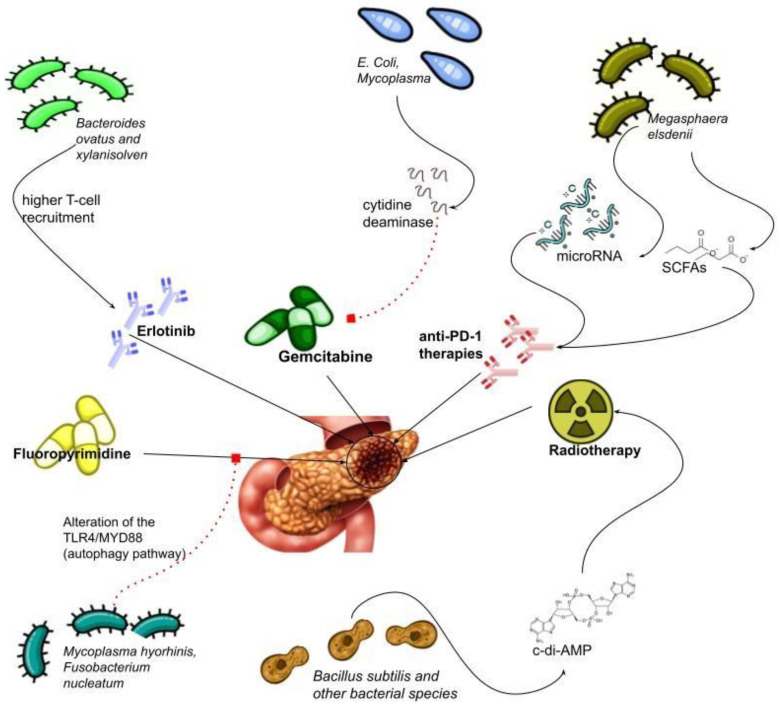
Possible effects of different bacterial strains and their metabolites on therapies of pancreatic cancer. Continuous lines indicate stimulation; dashed lines indicate inhibition.

**Table 1 biomedicines-11-00157-t001:** A panoramic of novel treatments in development for PC.

Pathway Inhibitors	Targeting DNA Damage Response (DDR)	Targeting Immune System	Targeting Tumor Metabolism	Targeting Tumor Stroma Fibrosis, and Extracellular Matrix
KRAS	The Poly (ADP-ribose) polymerase (PARP) Inhibitor	Immune Checkpoint Inhibitors (PD-L1/PD-1, CTLA-4)	Targeting Tricarboxylic acid (TCA) enzymes	Targeting Hyaluronic Acid
Neurotrophic Tyrosine Receptor Kinase (NTRK) fusion		Vaccines		Targeting Immune Cells/Signals inside the stroma
Neuregulin-1 gene (NRG1) Fusion		CAR-t Cells Transfusion CD-40, IL-10 (failed to provide benefit)		

**Table 2 biomedicines-11-00157-t002:** Principal microbiological fundings in pancreatic cancer tissue.

Tumour-Associated Microbiota in Pancreatic Cancer		
	Increased Presence	Reduced Presence
Phylum	*Euryarchaeota*	*Firmicutes* and *Proteobacteria*
Genus	- *Porphyromonas, Prevotella, Bifidobacterium and Synergistetes* - *Acinetobacter, Sphingopyxis* and *Pseudomonas* (KRAS mutations) - *Malassezia* and *Alternaria* (Mycobiome)	- *Lactobacillus* - *Pseudoxanthomonas, Saccharopolyspora, Streptomyces* (worse prognosis)
Species	*Fusobacterium nucleatum*, *Beutenbergia cavernae* DSM 12333, *Mycoplasma hypopneumoniae*, *Tolypothrix sp. PCC 7601*, *Acidovorax ebreuus TPSY*, *Agrobacterium radiobacter* K84, *Aggregatibacter aphorophilus* NJ8700, *Shigella sonnei* Ss046, *Salmonella enterica*, *Citrobacter freundii*	- *Saccharomonospora Viridis* DSM 43017 - *Alkalihalobacillus clausii* (worse prognosis)

## Data Availability

Not applicable.

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
