# Peer review of "Pancreatic Cancer Resistance to Treatment: The Role of Microbiota"

_biomedicines, 2023, doi:10.3390/biomedicines11010157_

Round 1

Reviewer 1 Report

In this manuscript, the authors presented the role of microbiota on pancreatic cancer resistance to treatment. Pancreatic cancer is one of the lethal cancers with late detection and resistance to treatment affecting the patients’ survival. Here, the authors explained the role of microbiota in influencing the treatment with evidence from animal and human studies. This information will be helpful for researchers working in pancreatic cancer and others interested in understanding the disease mechanism.

The authors can improve the overall presentation of the manuscript. The following are some minor comments to improve the manuscript.

3.1. should have been written as 1.1. for “Therapies for pancreatic cancer” section. The paragraph following that appears to be an outline for the authors rather than for the readers.

NCCN was written as NCCT in L97 and no expansion was provided for it.

PDAC was written as PDCA in many places and please fix those.

The authors mentioned several studies in L127 and provided a single reference for that statement.

Please change the statement “bacterial DNA appears to be overexpressed” in L145 as this is not appropriate.

The expansion for FMT was given in L266 instead of L182 and similarly for CA 19-9 in L198 instead of L68.

Cytidine deaminase is abbreviated as CDD instead of CDA.

I believe Fusobacterium was written as Fusobacterioma in L184.

In L141, gingivalis was written as gengivalis.

In L212, the term “human models” is not appropriate in that sentence.

In L357, abbreviation DDC was given without expansion.

Author Response

First, we would like to thank the reviewers for taking the time to evaluate our manuscript and for the valuable suggestions that allowed us to improve the paper.

Reviewer 1

1) 3.1. should have been written as 1.1. for “Therapies for pancreatic cancer” section. The paragraph following that appears to be an outline for the authors rather than for the readers.

Thank you very much for these suggestions. We have corrected the numbering of the paragraph and deleted the first sentence in paragraph 1.1

2) NCCN was written as NCCT in L97 and no expansion was provided for it.

We have corrected the acronym as suggested by the reviewer.

3) PDAC was written as PDCA in many places and please fix those.

We have corrected the acronym as suggested by the reviewer throughout the text

4) The authors mentioned several studies in L127 and provided a single reference for that statement.

We have fixed the inconsistancy

5) Please change the statement “bacterial DNA appears to be overexpressed” in L145 as this is not appropriate

The sentence has been changed as suggested by the reviewer

6) The expansion for FMT was given in L266 instead of L182 and similarly for CA 19-9 in L198 instead of L68.

We have corrected the problem as suggested by the reviewer throughout the text

7) Cytidine deaminase is abbreviated as CDD instead of CDA.

We have corrected the acronym as suggested by the reviewer throughout the text

8) I believe Fusobacterium was written as Fusobacterioma in L184.

We have fixed the typo

9) In L141, gingivalis was written as gengivalis.

We have fixed the typo

10) In L212, the term “human models” is not appropriate in that sentence.

The word model has been deleted

11) In L357, abbreviation DDC was given without expansion.

DDC was a typo for CDD now corrected in CDA as explained in point 7

Reviewer 2 Report

it is a well written and interesting review, however some minor points should be revised

1. plese check at section 3.1 the lines 77-79 seems more instructions and no related to the manuscript

2. the potential role of probiotic and fecal transplantation as therapeutic approaches should also discussed

Author Response

First, we would like to thank the reviewers for taking the time to evaluate our manuscript and for the valuable suggestions that allowed us to improve the paper.

Reviewer 2

it is a well written and interesting review, however some minor points should be revised

  1. please check at section 3.1 the lines 77-79 seems more instructions and no related to the manuscript

Thank you very much for these suggestions. We have eleted the first sentence in paragraph 1.1

  1. the potential role of probiotic and fecal transplantation as therapeutic approaches should also discussed

We have included a brief summary of the therapeutic use of FMT and probiotics in pancreatic cancer in the conclusions
